# Lung Tumor Cells with Different Tn Antigen Expression Present Distinctive Immunomodulatory Properties

**DOI:** 10.3390/ijms231912047

**Published:** 2022-10-10

**Authors:** Valeria da Costa, Karina V. Mariño, Santiago A. Rodríguez-Zraquia, María Florencia Festari, Pablo Lores, Monique Costa, Mercedes Landeira, Gabriel A. Rabinovich, Sandra J. van Vliet, Teresa Freire

**Affiliations:** 1Laboratorio de Inmunomodulación y Vacunas, Departamento de Inmunobiología, Facultad de Medicina, Universidad de La República, Montevideo 11800, Uruguay; 2Laboratorio de Glicómica Funcional y Molecular, Instituto de Biología y Medicina Experimental (IBYME), Consejo Nacional de Investigaciones Científicas y Técnicas (CONICET), Buenos Aires 1428, Argentina; 3Laboratorio de Glicomedicina, Instituto de Biología y Medicina Experimental (IBYME), Consejo Nacional de Investigaciones Científicas y Técnicas (CONICET), Buenos Aires 1428, Argentina; 4Department of Molecular Cell Biology and Immunology, Amsterdam UMC, Location Vrije Universiteit Amsterdam, De Boelelaan 1117, 1081 HV Amsterdam, The Netherlands; 5Amsterdam Institute for Infection and Immunity, Cancer Immunology, 1081 HV Amsterdam, The Netherlands; 6Cancer Center Amsterdam, Cancer Biology and Immunology, 1081 HV Amsterdam, The Netherlands

**Keywords:** lung cancer, Tn antigen, Macrophage Galactose-type lectin, *O*-glycosylation, dendritic cells

## Abstract

Lung cancer is the first leading cause of cancer-related deaths in the world. Aberrant glycosylation in lung tumors leads to the expression of tumor-associated carbohydrate structures, such as the Tn antigen, consisting of *N*-acetyl-galactosamine (GalNAc) linked to a serine or threonine residue in proteins (α-GalNAc-O-Ser/Thr). The Tn antigen can be recognized by the Macrophage Galactose/GalNAc lectin (MGL), which mediates various immune regulatory and tolerogenic functions, mainly by reprogramming the maturation of function of dendritic cells (DCs). In this work, we generated two different Tn-expressing variants from the Lewis-type lung murine cancer cell line LL/2, which showed different alterations in the *O*-glycosylation pathways that influenced the interaction with mouse MGL2 and the immunomodulatory properties of DCs. Thus, the identification of the biological programs triggered by Tn^+^ cancer cells might contribute to an improved understanding of the molecular mechanisms elicited by MGL-dependent immune regulatory circuits.

## 1. Introduction

Lung cancer is the first world leading cause of cancer-related deaths for both men and women [1,2]. Aberrant glycosylation is a hallmark of epithelial tumors [3], including lung cancer [4,5]. Altered glycosylation pathways of tumors lead to the expression of tumor-associated carbohydrate antigens [3], such as the Tn antigen, which is highly expressed by adenocarcinomas [6]. Tn antigen is associated with cancer aggressiveness, poor prognosis, and metastasis [6,7]. These changes can also be detected in circulating tumor-derived glycoproteins from non-small cell lung cancer (NSCLC) patients, where an increase in Tn-expressing glycoproteins is detected [4]. Tn antigen consists of *N*-acetyl-galactosamine (GalNAc) linked to a serine or threonine residue in proteins (α-GalNAc-O-Ser/Thr) and results from the blockade of the mucin-type *O*-glycosylation pathways caused by defects in essential proteins or enzymes involved in its synthesis, like polypeptide-GalNAc-transferases (ppGalNAc-T), T-synthase, and its associated chaperone, core 1 GalT1 chaperone 1 (Cosmc) [8,9,10,11,12,13,14].

The tumor cell surface glycan signatures can be recognized by different glycan-binding proteins, including Siglecs, Galectins, and C-type lectin receptors (CLRs), which are present on myeloid antigen presenting cells [15,16]. In particular, the Macrophage Galactose/GalNAc lectin (MGL; CD301) binds terminal GalNAc residues present on pathogens, transformed cells, or endogenous ligands in a calcium-dependent manner and mediates various immune and homoeostatic functions [17]. Two similar MGL isoforms (mMGL1 and mMGL2) have been identified in mice. However, only mMGL2 specifically recognizes GalNAc residues, including the Tn antigen [18,19]. Both MGL expression and function have been associated with impaired or tolerogenic immune responses, mainly through modulation of the maturation and function of dendritic cells (DCs) or macrophages [20,21,22] and the secretion of anti-inflammatory cytokines, such as interleukin 10 (IL-10) [23].

Clustered regularly interspaced short palindromic repeats (CRISPR) and CRISPR-associated protein 9 (Cas9) gene editing is a helpful strategy for studying the function of proteins [24,25]. It has been used for understanding the role of Tn antigen in tumor development by knocking out *Cosmc*, a strategy that efficiently induces expression of the Tn antigen [8,9,26,27]. Experiments using both human and mouse cancer *Cosmc* knock-out cell lines have demonstrated that the Tn antigen is involved in metastasis, angiogenesis, and immune escape [7,28,29,30,31].

In this study, we aimed to study the role of aberrant *O*-glycosylation in the interaction by MGL2 and by DCs, the main orchestrator of adapting immunity. To this end, we generated two different Tn-expressing variants from the Lewis-type lung murine cancer cell line LL/2 by knocking out *Cosmc* using CRISPR/Cas9 gene editing. We characterized their glycophenotype and immunomodulatory properties on DCs and demonstrate that both Tn^+^ tumor cell variants exhibit different glycan structures, which differentially modulate MGL2 binding and immunological effects on DCs. Thus, the deletion of Cosmc function can generate distinct Tn^+^ cancer cells with critical modification of *O*-glycosylation pathways and different capacities to regulate immune responses.

## 2. Results and Discussion

### 2.1. Incomplete O-Glycosylation Alters the Glycophenotype of LL/2 Cells

Tn-expressing lung tumor cells. Tn^+^ cells were generated from the murine lung Lewis adenocarcinoma cell line (LL/2-WT) by knocking out *Cosmc* using CRISPR/Cas9 gene editing. Tn^+^ cells were selected based on recognition of the anti-Tn monoclonal antibody (mAb) 83D4, since this antibody binds both trimeric and dimeric Tn forms on cancer cells [32]. The selection procedure was performed either by enriching in 83D4-reactive LL/2 cells (H12) or by single cell deposition (F9) by flow cytometry. Of note, LL/2-Tn^+^-H12 cells seemed to be more adherent than LL/2-Tn^+^-F9 and LL/2-WT cells since they exhibited a rod-like shape, not rounded (Figure 1A). Flow cytometry analysis revealed both Tn^+^ cell lines have a significant increase in Tn expression when stained with by the anti-Tn 83D4 mAb, compared to the parental line, being LL/2-Tn^+^-F9 a more homogeneous population regarding Tn expression as compared to LL/2-Tn^+^-H12 (Figure 1B), although no significant differences in the median fluorescence intensity were found between the two cell variants (Figure 1C). In addition, T-synthase activity, the enzyme responsible for elongating the Tn antigen and the formation of core 1 (Figure 1D), was abrogated in both Tn^+^ cells (Figure 1E), thus confirming the interruption of the step catalyzed by this enzyme.

Interestingly, LL/2-Tn^+^-F9 proliferated slightly slower than LL/2-Tn^+^-H12 and LL/2-WT cells (Figure 2A). However, the two LL/2-Tn^+^ cell variants and parental cell line did not differ in their migration capacity or cell colony formation in vitro (Figure 2B,C).

In order to investigate surface glycan structures, we performed flow cytometry assays using biotinylated plant lectins with different glycan specificity Appendix A). These assays revealed variations in the glycophenotype of Tn^+^ and WT LL/2 cells (Figure 3). As expected, *Vicia villosa* lectin (VVL, specific for Tn antigen) showed significantly higher recognition levels for LL/2-Tn^+^-H12 and LL/2-Tn^+^-F9 cells when compared to WT cells, indicating the presence of higher levels of Tn on these cells (Figure 3B). Jacalin (a lectin that can recognize Tn, core 1 and sialylated core 1 antigens) [33] presented a similar binding pattern to VVL, thus validating the higher Tn antigen levels in these cells (Figure 3C). Likewise, *Helix pomatia* agglutinin (HPA, specific for GalNAc) similarly recognized both Tn^+^ cell variants, although a slight but significant decrease in the binding pattern was observed in the case of LL/2-Tn^+^-H12 as compared to LL/2-Tn^+^-F9 tumor cells (Figure 3D). Moreover, binding of *Soybean agglutinin* (SBA) and *Dolichos biflorus agglutinin* (DBA) was significantly lower for LL/2-Tn^+^-H12 and LL/2-Tn^+^-F9 cells, respectively, than LL/2-WT cells (Figure 3E,F). Finally, although no changes were observed in recognition by *Maackia amurensis* lectin II (MAL II), a plant lectin specific for α(2-3)-linked sialic acid, between Tn^+^ or WT cells (Figure 3G), a decrease in binding of *Sambuccus nigra agglutinin* (SNA, specific for α(2-3)-linked sialic acid) to LL/2-Tn^+^-H12 cells, but not LL/2-Tn^+^-F9 cells was observed, revealing lower proportion of α(2-6)-linked sialic acid with respect to WT cells (Figure 3H).

Altogether, these results demonstrate that knocking out *Cosmc* via CRISPR/Cas9 gene editing in LL/2 cells efficiently generates Tn^+^ tumor cell variants with high levels of Tn antigen, as shown by VVL binding. However, intrinsic differences regarding their glycosylation profile could be detected. While the LL/2-Tn^+^-H12 cell variant strongly reacts with Jacalin, it also exhibits lower α(2-6)-sialylation with respect to LL/2-Tn^+^-F9 and WT cells, suggesting that α(2-6)-sialylation, including that displayed by the Tn antigen might be reduced in LL/2-Tn^+^-H12 cells. It was previously shown that Jacalin could not bind to sialyl-Tn-attached peptides as efficiently as to Tn antigen, as the C6-OH of GalNAc is substituted [33]. The different profiles obtained with DBL and SBA could also point to other alterations in the cellular glycome, as these lectins have a broader scope of recognition, including structures with terminal βGalNAc for both lectins, and βGal for SBA [34]. It is worth noting that knocking out *Cosmc* by CRISPR/Cas9 gene editing has been a strategy widely used in the last two decades to study the role of aberrant glycosylation in tumor development [7,28,35,36], but no exhaustive glycophenotype characterization of *Cosmc* KO cells has been performed. Thus, mutation of *Cosmc* might induce compensatory mechanisms, including transcriptional or epigenetic regulation of glycosyltransferases. Moreover, enzymatic dysregulation could be the result of signaling pathways potentially modulated by aberrant *O*-glycosylation, as recently demonstrated in gastric cancer cells [37].

### 2.2. MGL2 Differentially Recognizes Tn^+^ LL/2 Cells

Next, considering their role in DC modulation, we analyzed the recognition of LL/2-Tn^+^-H12 and -F9 cells by MGL1 and MGL2 CLRs. Flow cytometry analyses using chimeric MGL1-Fc and MGL2-Fc demonstrated that MGL2 recognized both Tn^+^ LL/2 cells, and this recognition was abrogated with EGTA (Figure 4A), while MGL1 displayed only minimal binding and likely not mediated by the carbohydrate-recognition domain since incubation with EGTA did not prevent this effect (Figure 4B). However, MGL2 preferentially recognized LL/2-Tn^+^-F9 compared to LL/2-Tn^+^-H12 Tn^+^ cells (Figure 4A). Interestingly, only LL/2-Tn^+^-F9 cells, but not LL/2-Tn^+^-H12, were able to interact with surface MGL2 recombinantly expressed on the surface of Chinese hamster ovary (CHO) cells, and this interaction was abrogated in the presence of EGTA (Figure 4C). These results were confirmed by lectin blotting, since MGL2 recognized >75 kDa components in LL/2-Tn^+^-F9 lysates that were only slightly recognized on LL/2-Tn^+^-H12 cell lysate (Figure 4D,E). Of note, MGL2 recognition of both Tn^+^ LL/2 cells was inhibited by VVL binding Appendix A, indicating that MGL2 interaction with glycoproteins of LL/2-Tn^+^ cells was mediated by the Tn antigen. Finally, to analyze whether MGL2 and the 83D4 anti-Tn mAb interact with the same Tn structures on the surface of Tn^+^ LL/2 cells, we carried out inhibition assays by flow cytometry, first incubating with the MGL2-Fc followed by staining with the 83D4 anti-Tn antibody. Results demonstrated that MGL2-Fc was only capable of inhibiting 83D4 anti-Tn mAb binding to LL/2-Tn^+^-H12, but not to LL/2-Tn^+^-F9 cells (Figure 4F).

Collectively, these results show that MGL2 interacts with both Tn^+^ LL/2 cells, but LL/2-Tn^+^-F9 is strongly recognized by both soluble and cell surface MGL2, indicating that the molecular presentation of the Tn antigen on the surface of the tumor cell might be playing a role in the interaction with MGL2. This result is also supported by the fact that the MGL2-Fc was able to inhibit the anti-Tn mAb binding to LL/2-Tn^+^-H12 but not to LL/2-Tn^+^-F9 cells. Previous studies have established that the binding of some anti-Tn antibodies could be affected by the density of Tn determinant or/and by the amino acid residues neighboring O-glycosylation sites [32,38]. Different binding patterns of these antibodies were also observed in three human cancer cell lines (MCF-7, LS174T and Jurkat) [38]. Thus, it is likely that MGL2 and the anti-Tn 83D4 mAb interact differently with both Tn^+^ cancer cells according to molecular presentation of the Tn antigen by these cells. Furthermore, we have recently shown that human MGL also recognizes part of the peptide backbone next to the Tn antigen [39]. Thus, the underlying protein carrying Tn antigen might also play a role in different binding of MGL2 to different Tn^+^ cell lines.

### 2.3. LL/2-Tn^+^-H12 and LL/2-Tn^+^-F9 Differ in Their Ability to Modulate Bone-Marrow-Derived DC (BMDC) Function

To analyze the capacity of Tn^+^ LL/2 tumor cells to modulate the maturation of DCs, we incubated BMDC (which express MGL2, Figure 5A) with soluble components derived from LL/2-Tn^+^-H12, LL/2-Tn^+^-F9 or WT cells present in serum-free conditioned medium (CM). Then, we evaluated the expression of costimulatory molecules on BMDCs (such as MHCII, CD80, and CD86) by flow cytometry (Figure 5B) and the secretion of pro- and anti-inflammatory cytokines (IL-12 and TNFα, IL-10) by specific sandwich ELISA (Figure 5C). A considerable increase in CD86 expression was detected when CD11c^+^ MGL2^+^ BMDCs were incubated with CM from LL/2-Tn^+^-F9 cells when compared to LL/2-WT cells. However, no significant differences were observed for CD80, or MHC class II expression on BMDCs incubated with CM from LL/2-Tn^+^-H12 or LL/2-WT tumor cells (Figure 5B).

Moreover, CM from LL/2-Tn^+^-H12 cells induced the secretion of higher levels of IL-10 as compared to CM obtained from LL/2-Tn^+^-F9 and WT tumor cells, while CM from LL/2-Tn^+^-F9 induced the secretion of higher levels of TNFα and IL-12 (Figure 5C). Finally, we observed a decrease in the IL-12/IL-10 ratio induced by soluble factors from LL/2-WT and LL/2-Tn^+^-H12 cells when compared to LL/2-Tn^+^-F9 cells and controls (Figure 5C). Furthermore, BMDCs incubated with LL/2-Tn^+^-H12 CM presented even lower levels of IL-12/IL-10 than LL/2-WT cells (Figure 5C). Altogether, these results indicate that, while LL/2-Tn^+^-F9 cells induce the maturation of pro-inflammatory-like BMDC, LL/2-Tn^+^-H12 cells favor a predominant anti-inflammatory effect on BMDC, by promoting the secretion of IL-10.

It was recently shown that Tn^+^ tumors foster an immunosuppressive microenvironment characterized by IL-10 expression and mediated by MGL2^+^ antigen-presenting cells [29], enhanced accumulation of myeloid-derived suppressor cells (MDSCs), and PD-L1^+^ macrophages [31] as well as reduced frequency of cytotoxic CD8^+^ T cells [28], indicating that MGL2 expression in the tumor associates with immune escape. In addition, both expression and function of MGL have been widely associated with tolerogenic and immunosuppressive responses. In fact, MGL is expressed mainly on immature or tolerogenic DCs [20], corticosteroid-cultured macrophages [21], alternative activated macrophages [22] and type 2 DCs [40,41]. Furthermore, MGL triggering dampens the immune response by inducing the production of the anti-inflammatory cytokine IL-10 by DCs [23], promoting the differentiation of Tregs [42], and inducing T cell apoptosis [22]. Last, tolerogenic DCs generated in the presence of different stimuli increase MGL expression [20,21], secrete high amounts of both IL-10 and TNFα, and instruct the differentiation of T cells towards a regulatory T cell phenotype in an IL-10 and TNFα-dependent manner [23,42]. Interestingly, LL/2-Tn^+^-H12 cells, but not LL/2-Tn^+^-F9, induced a lower IL-12/IL-10 ratio and TNFα production by BMDC than LL/2-WT and BMDC without tumor cells (Figure 4C), suggesting that LL/2-Tn^+^-H12 cells, but not LL/2-Tn^+^-F9, are able to regulate the maturation of DCs into an immunoregulatory state. IL-10 possesses potent anti-inflammatory properties and plays a central role in limiting anti-tumor immune response. Indeed, IL-10, together with TGFβ, is one of the major effectors of regulatory T cell development [43,44,45]. In contrast, IL-12 can induce a Th1 immune response and the induction of cytotoxic CD8^+^ T cells and activation of natural killer cells [45,46]. On the other hand, TNFα is a pleiotropic cytokine involved in immune defense against cancer by promoting inflammation and apoptosis of cancer cells [47]. Furthermore, TNFα can also induce the shedding of relevant molecules present in leukocytes, such as CD45 and CD30, and could eventually modulate the anti-tumor immune response [47]. Therefore, our results indicate that different Tn^+^ tumor cells can induce both anti- and pro-inflammatory properties in DCs, indicating that either the expression of the Tn antigen or other glycan structures by tumor cells are critical in promoting immune escape. Indeed, Tn-based proteins or peptides may also induce a pro-inflammatory immune response through the production of specific antibodies that recognize tumor cells, in a process where the peptide backbone, the Tn density, and the adjuvant used might be crucial to determine the nature and magnitude of the ongoing immune response [41,48].

In addition, the changes observed in the expression of activation molecules and cytokine production by BMDC suggest that the soluble components secreted by both Tn^+^ cells differ from each other. This could be a result of the differences observed in Tn expression by both cell lines or could be associated with the role of the specific proteins that carry the Tn antigen that interacts with MGL2, or with the Tn antigen density within a polypeptide, which could also affect MGL2 recognition and therefore modulate the BMDC response. Furthermore, the described differences regarding the glycans alterations in tumor Tn^+^ cells could also suggest the engagement of other CLRs expressed by the BMDC, which could be triggered by different glycosidic antigens in the tumor cells. Finally, this alteration in glycan expression could also suggest modifications occurring at a transcriptomic level within the tumor cells, which could lead to differential protein secretion, and therefore, engage other receptors in the BMDC response. In summary, the two different approaches were successful in the generation of high-expressing Tn-expressing tumor cells, although they presented significant differences in other glycan expression and in their capacity to modulate DCs, which could eventually trigger different immune responses against cancer.

## 3. Materials and Methods

### 3.1. Mice

C57BL/6 mice were purchased from DILAVE Laboratories (Uruguay). Mouse handling and experiments were carried out in compliance with institutional guidelines and regulations from the National Committee on Animal Research (CNEA, Uruguay) and approved by the Universidad de la República’s Committee on Animal Research (CHEA Protocol N° 070153-000811-19).

### 3.2. Generation of Tn^+^ Cells

The murine lung cancer cell line LL/2 was purchased from ATCC (Manassas, VA, USA) in 2015 and cultured for up to 8–10 passages. *Mycoplasma* testing was performed by PCR. Transfection of LL/2 cells was carried out by CRISPR guide targeting to *Cosmc* exon 2-690s (GCGGTCTGCCTGAAATACGC) cloned in pBS-U6sg plasmid (Tacgene, Paris, France), required for folding the T-synthase. Cells were incubated with the anti-Tn monoclonal antibody 83D4 [32], followed by an anti-mouse IgM-FITC, and Tn^+^ cells were sorted with BD FACSAria™ Fusion (BD Biosciences, Franklin Lakes, NJ, USA). The first Tn^+^ cell variant was obtained by enriching Tn^+^ cells (LL/2-Tn^+^-H12), while the second Tn^+^ cell clone was obtained by single-cell deposition (LL/2Tn^+^-F9). The mutations in *Cosmc* in the selected cell clones were further verified by Sanger sequencing using the following primers: Cosmc-F: 5′-TGCTTTCACTTGCCACTTTG-3′ and Cosmc-R: 5′-TTCCTCCATCCACACTCACA-3′. A substitution of a G for A was found in both Tn^+^ cell lines in nucleotide position 3171, as follows: 5′-AAAGAC[*G/A*]ATATCT-3′. This mutation results in the substitution D49N in the protein sequence. Abrogation of T-synthase activity was confirmed as described previously [49]. This assay is specific to the activity of this enzyme since previous work demonstrated that deficiencies in T-synthase activity by this method correlate with mutations in the *Cosmc* gene and the introduction of wide type *Cosmc* into these cells restored the T-synthase activity [49].

### 3.3. SDS-PAGE and Western Blot

Protein lysates were run on 8% SDS polyacrylamide gels and transferred to a nitrocellulose membrane (GE Healthcare, Buc, France). Membranes were blocked overnight with Roti-Block (Carl Roth, Karlsruhe, Germany) and incubated with recombinant murine MGL2-Fc in TSM (20 mM Tris-HCl, 150 mM NaCl, 2 mM MgCl_2_, 1 mM CaCl_2_, pH 7.4), followed by incubation with a peroxidase-labeled anti-human IgG-Fc antibody (Jackson Immunoresearch Laboratories, West Grove, PA, USA) or peroxidase-labeled streptavidin. Alternatively, for the inhibition assay, the membranes were pre-incubated with biotinylated Isolectin B4 of *Vicia villosa* (VVL) prior to the MGL2-Fc staining.

### 3.4. Glycophenotyping of Tumor Cells

Wild type (WT) and Tn^+^ LL/2 tumor cells were stained with anti-Tn antibody 83D4 or the following lectins (Vector Labs, Newark, CA, USA): *Helix pomatia* lectin (HPA), VVL, *Sambucus Nigra* lectin (SNA), *Maackia amurensis* II lectin (MAL II),), *Dolichos biflorus* agglutinin (DBA) and Soybean agglutinins (SBA) in PBS containing 2% BSA, 0.02% NaN_3_, or MGL1-Fc and MGL2-Fc in Hanks’ Balanced Salt Solution (HBSS) with 0.5% BSA, for 30 min at 4 °C. Specific glycan epitopes recognized by each lectin are described in Appendix A. In order to verify carbohydrate recognition, EGTA (10 mM) was added to MGL2-Fc staining. Cells were incubated with streptavidin-PE, FITC-conjugated anti-mouse IgM, or APC-conjugated anti-human IgG. Finally, cells were fixed with formaldehyde 1% in PBS. For lectin binding inhibition assays, LL/2 cells were pre-incubated with MGL2-Fc prior 83D4 incubation and staining.

### 3.5. Cell Surface MGL2 Recognition of Tumor Cells

MGL2-expressing CHO produced as previously described [50] were stained with calcein-AM and incubated in 96-well plates with the wild type (WT) and Tn^+^ LL/2 tumor cells. In order to verify carbohydrate recognition, EGTA (10 mM) was added. After 5 min, the non-adherent cells were washed, and the remaining cells were lysed in 50 mM Tris 0.1% SDS. Fluorescence was measured at λex 480 nm, λem 530 nm.

### 3.6. BMDC Functional Assays

BMDCs were generated from bone marrow precursors supplemented with GM-CSF (20 ηg/mL, Peprotech, Cranbury, NJ, USA). BMDCs (3 × 10^5^/well) were incubated at 37 °C and 5% CO_2_ in 96-well plates with soluble factors derived from both Tn^+^ or Tn^-^ tumors and LL/2-Tn^+^ or LL/2-Tn^-^ cells overnight at 37 °C. BMDCs were centrifuged at 1500 rpm for 5 min at 4 °C, and supernatants were then collected. Cytokine (IL-12/23p40, IL-10, and TNFα) levels were tested on culture supernatants by cytokine-specific sandwich ELISA assays (BD Bioscience, Franklin Lakes, NJ, USA). Soluble factors in CM from LL/2-Tn^+^-F9, -H12, or LL/2-WT cells corresponded to the culture medium derived from an 18 h culture of 5 × 10^6^ tumor cells in 100 mm Petri dishes.

### 3.7. Statistical Analyses

Statistical analyses were performed with GraphPad Prism 6.01 (San Diego, CA, USA). The Student’s unpaired *t* test was used to determine the significance of the differences. Asterisks indicate statistically significant differences with * *p* < 0.05, ** *p* < 0.01, *** *p* < 0.001. Data are representative of at least two individual experiments.

## 4. Conclusions

In conclusion, this report demonstrates that Tn^+^ tumor cells obtained by knocking out *Cosmc* exhibit unique alterations in O-glycan profiles, thus influencing interaction with MGL2 and the effects on DCs. Thus, characterization of glycosylation signatures of Tn^+^ tumor cells along with the identification of the proteins carrying the Tn antigen, may be critical to elucidate variations in the intrinsic glycosylation machinery of tumor cells that could modulate their interactions with MGL-expressing leukocytes through the Tn antigen and reprogram anti-tumor immune responses.

## Figures and Tables

**Figure 1 ijms-23-12047-f001:**
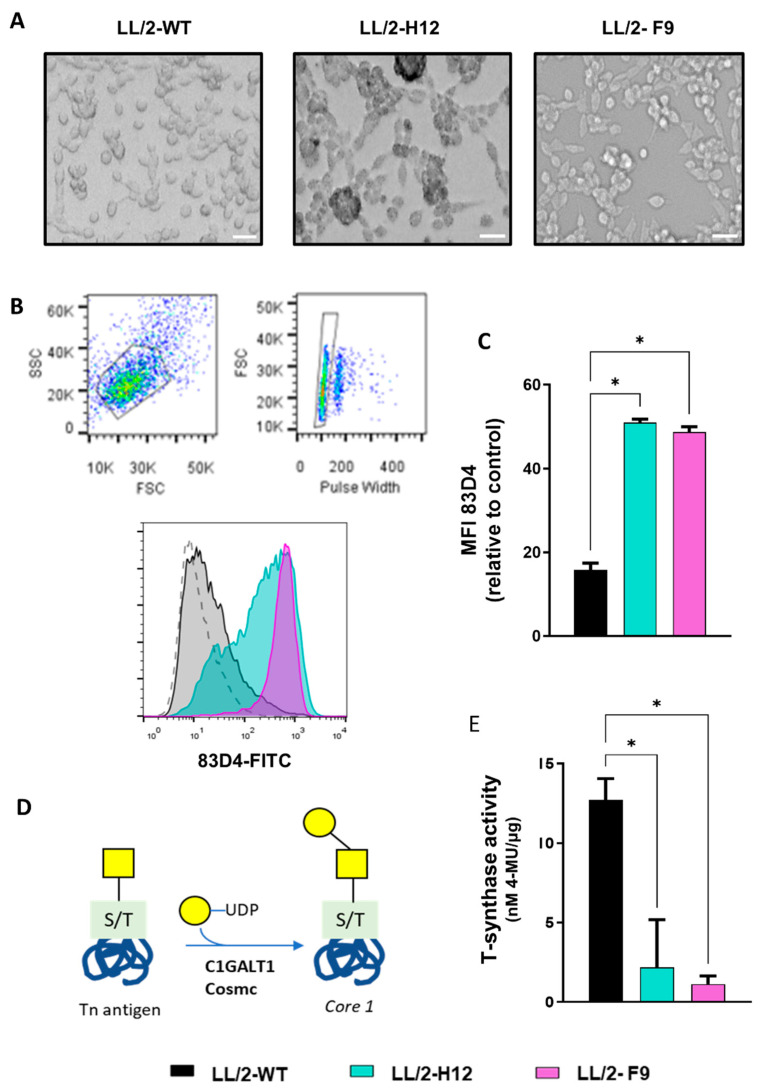
Generation of Tn^+^ LL/2 cells. (**A**) Morphology of LL/2 WT and Tn^+^ cells (LL/2- Tn^+^-H12 and LL/2-Tn^+^-F9). The white bar represents 20 µm. (**B**) Expression of Tn antigen by flow cytometry in LL/2 tumor cells. Representing gating strategy and histogram of 83D4 staining on LL/2-WT (black), LL/2-Tn^+^-H12 (green), and LL/2-Tn^+^-F9 (purple) cells. The dotted line represents the non-stained negative control. (**C**) Median fluorescence intensity (MFI) of 83D4 staining normalized to negative control incubating cells only with secondary antibody FITC-conjugated (dotted histogram). (**D**) Representation of the Tn elongation to core 1 by the core 1 synthase (C1GALT1, T-synthase) enzyme together with the Cosmc chaperone. (**E**) T-synthase activity was measured as nM of 4-Methylumbelliferone (4-MU) per µg of total protein. Asterisks represent significant differences (*p* < 0.05).

**Figure 2 ijms-23-12047-f002:**
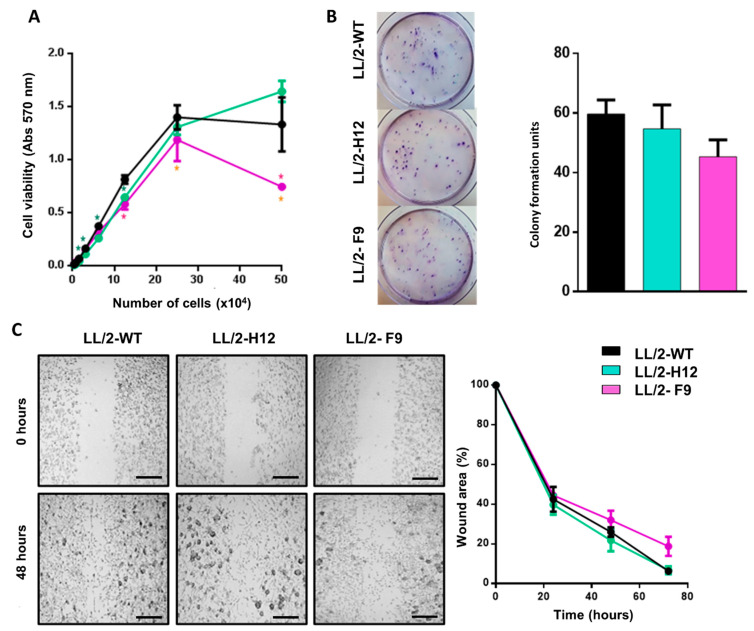
Tn^+^ LL/2 cell variants present similar proliferation and migration properties to parental cell line. (**A**) Cell growth evaluated by viability with MTT. (**B**) Clonogenic ability of cells measured by colony forming assay. Number of colony formation units was counted manually per plate. (**C**) Cell migration by wound healing assay. Measures were taken at 0, 24, 48, and 72 h, and the area was determined using ImageJ. Wound area is represented as percentage or area relative to time point 0. was calculated. The images represent at time 0 and time 48 h. The black bar represents 100 µm Asterisks represent significant differences (*p* < 0.05).

**Figure 3 ijms-23-12047-f003:**
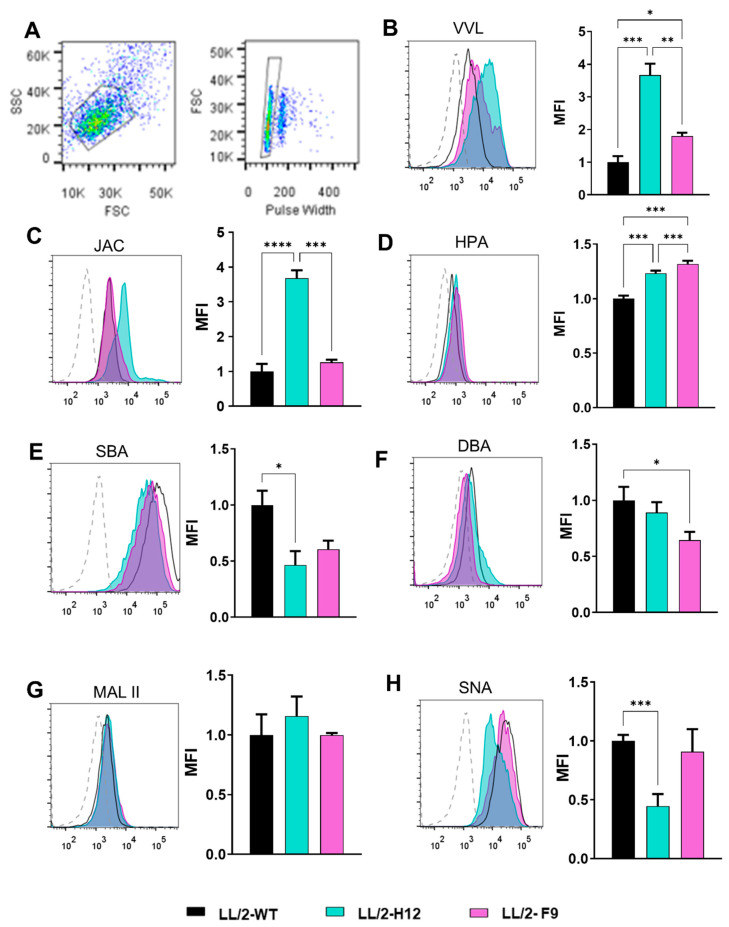
Surface glycan motifs of Tn-bearing LL/2 and WT cells determined by lectin recognition. LL/2 cells were stained with plant lectins and analyzed by flow cytometry. (**A**) Representative gating strategy. B-H) Histogram and median fluorescence intensity (MFI) are shown for LL/2-Tn^+^-H12 (green) and LL/2-Tn^+^-F9 (pink) relative to WT LL/2 cells for VVL (**B**), JAC (**C**), HPA (**D**), SBA (**E**), DBA (**F**), MAL II (**G**) and SNA (**H**). Dotted histograms represent the control condition consisting of cells incubated only in the presence of streptavidin-FITC. The specificity of lectins is shown in Appendix A. Asterisks represent significant differences (* *p* < 0.05, ** *p* < 0.01, *** *p* < 0.001, **** *p* < 0.00001).

**Figure 4 ijms-23-12047-f004:**
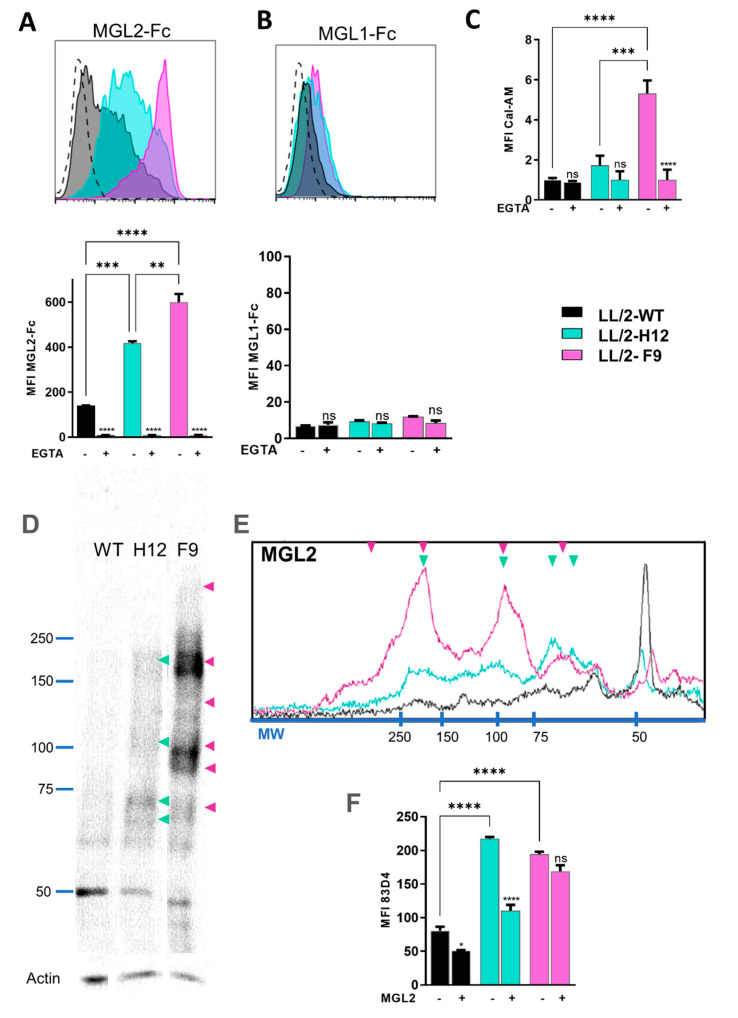
MGL2 differently interacts with LL/2-Tn^+^ cell variants. (**A,B**) Recognition of LL/2 tumor cells with soluble MGL2-Fc (**A**) or MGL1-Fc (**B**) analyzed by flow cytometry. Bottom, Median fluorescence intensity (MFI), in the presence or absence of EGTA. (**C**) Binding of MGL2^+^ CHO cells to the LL/2 cell variants by solid phase assay. (**D**) Lectin blot analysis of whole cell lysates, membranes using MGL2-Fc. Green and violet arrows represent components recognized in LL/2-Tn^+^-H12 and LL/2-Tn^+^-F9 cells, respectively. (**E**) Densitometry of MGL2-Fc recognition of cell lysates by lectin blotting. (**F**) Anti-Tn 83D4 mAb staining of LL/2 cell lines, with or without pre-incubation with MGL2-Fc. Asterisks represent significant differences (* *p* < 0.05, ** *p* < 0.01, *** *p* < 0.001, **** *p* < 0.00001); ns: non-significant.

**Figure 5 ijms-23-12047-f005:**
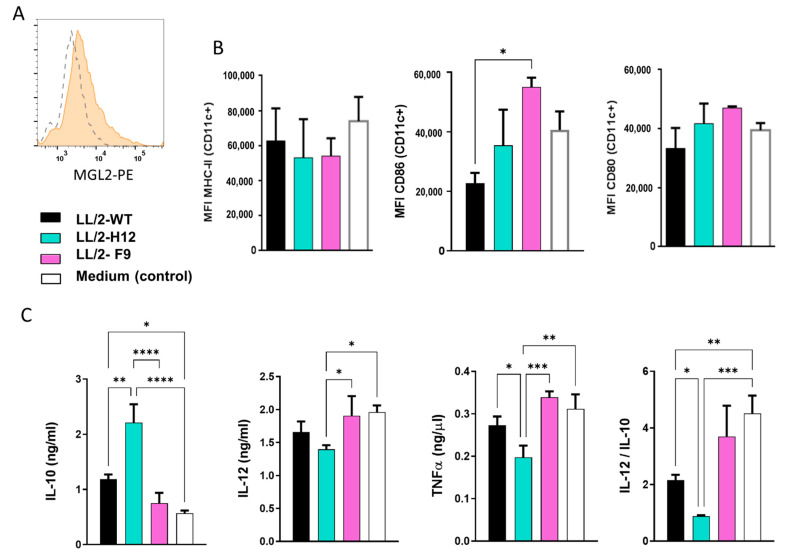
LL/2 derived-conditioned medium from Tn^+^ cells differ in their ability to modulate BMDC function. BMDCs were incubated overnight with CM derived from LL/2-WT, Tn^+^-H12, and Tn^+^-F9 cells. (**A**) MGL2 expression on BMDC by flow cytometry. (**B**) Expression of MHC II, CD86, and CD80 on CD11c^+^ MGL2^+^ BMDCs cultured with CM from LL/2 WT, Tn^+^-H12, and Tn^+^-F9 cells by flow cytometry. MFI indicates medium fluorescence intensity. (**C**) Cytokines (IL-10, IL-12, and TNF-α) were determined on culture supernatants by ELISA. Left, IL-12 to IL-10 ratio. Asterisks represents significant differences (* *p* < 0.05, ** *p* < 0.01, *** *p* < 0.001, **** *p* < 0.00001).

## Data Availability

Not applicable.

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
