# Peer review of "Lung Tumor Cells with Different Tn Antigen Expression Present Distinctive Immunomodulatory Properties"

_ijms, 2022, doi:10.3390/ijms231912047_

Round 1
Reviewer 1 Report
The authors demonstrated that Tn antigen tumor cells exhibited unique alterations in O-glycan profiles, which further influence interaction with MGL2 and DC response. The work is of great importance and well-deigned. The data and results adequately support the conclusion. I only found few minor issues:
Page 2, line 44, please give the full name of Cosmc.
Page 2, line 56, please give the full name of IL-10.
Page 2, line 75, the right bracket is missing.
Author Response
Modifications suggested by the reviewer were included. Please see the revised version of the manuscript in red.
Reviewer 2 Report
this paper needs numerous changes so that the paper is not as confused and unclear as in this version
changes are necessary in many parts of the text and findings
1. the introduction is too crowded with data about glycans, and there is no significant connection between what the authors write and what is the goal of the work. Why all that was investigated is not very clearly written.???
2. in the material and method section, add the country of manufacture for certain reagents, especially cytokines, because the country of manufacture is not specified in that section, while the country of manufacture is specified in some other sections. Everything needs to be uniform.
3. at the end of the part of the text on measuring cytokines, add a reference that shows the method of measurement and add, as previously shown for cytokine production, Multiomic analysis of cytokines in immuno-oncology. Expert Rev Proteomics. 2020 Sep;17(9):663-674.
4. the first sentence in the results and discussion section begins with the objectives of the work, which should be in the introductory part of the text, while the findings should be in the results.
5. so change the introduction and supplement that part with clear objectives of the work.
6. I think that it is much better to write the results section in the paper, as in most other papers, and especially the discussion, because that way the confusion that exists in this text is avoided.
7. For the flow cytometry results in Figure 1b, it would have been much clearer if the cell gating strategy was attached immediately with the results and not in the supp figure.
8. In Figure 2c, 2d, it is not clearly indicated what the control is. Somewhere it says control in the text, but in the pictures it doesn't say control in the legend, but only the name of the cells, so it's confusing.
9. The sentence describing the flow cytometry findings is not clearly written: Flow cytometry analysis revealed a more homogeneous recognition of LL/2-Tn+ -F9 by the anti-Tn D4 antibody as compared to LL/2-Tn+ H12 (Figure 1B), although no significant differences in the median fluorescence intensity 85 were found between the two cell variants (Figure 1C).
10. Immediately below it is written that the enzymes showed different activity without describing how and in what way a computer-drawn picture is already presented and it is not the result.
11. After that, the authors write that the findings that show the difference are in the Supplementary Figure. Everything is completely confused
12. in the discussion regarding tumor suppression of cells in the tumor environment, cite references that show that numerous mediators and cytokines induce suppression and changes in membrane antigen expression that have been shown in various other models:
a) The role of cytokines in the regulation of NK cells in the tumor environment. Cytokine. 2019 May;117:30-40.
b) among many cytokines, TNF directly induces changes in the antigenic expression of individual cells, which has been demonstrated through the mechanism of tearing off tumor antigens and induction of apoptosis.
TNF-α induced apoptosis is accompanied by rapid CD30 and slower CD45 shedding from K-562 cells. J Membr Biol. 2011, 239(3):115-22.
13. In the legend and results in Figure 2, the name Glycophenotype is imprecise and general, it should be changed and named professionally. Somewhere in the text, one is mentioned, in the legends another, but it is not clearly written in the essence that it is the expression of markers on the surface of the cells.
14. In Figure 2, the legends also do not say what the control is, but it is only assumed that the first column is the control population of cells.
15. It would be good if the original flow cytometry data were first shown in Figure 2, and only then the bars with immunophenotype values
16. In the discussion and results, it is not written at all why CD86 was analyzed and for what reason and what its analysis means, but it is left to the reader to guess what the researchers wanted to say...
17. the variation in cytokine production in Figure 4 is different and this is not explained at all, which is why the changes in the concentration of TNF, which is proinflammatory and for which a reference was previously added that should be inserted, as well as for IL10, which is an inhibitory cytokine, are only repeated findings that can be seen from the picture
Author Response
- the introduction is too crowded with data about glycans, and there is no significant connection between what the authors write and what is the goal of the work. Why all that was investigated is not very clearly written.???
The introduction has several paragraphs
- lung cancer and the importance of glycoantigens in its growth.
- C-type lectin receptors and MGL2
- Use of Crispr technology
- A final paragraph indicating the objective of the study and the obtained results and conclusions.
We are convinced, that the information in this section is all relevant. However, to follow the reviewer´s advice, we have included a sentence explaining the goal of the study (lines 65-66).
In this study we aimed at studying the role aberrant O-glycosylation in the interaction by MGL2 and by dendritic cells, the main orchestrator of adapting immunity. To this end,
We would also like to point out that this paper has been submitted to the special Issue “New insights on roles of glycoconjugates in Health and disease” covers topics related with glycosylation and it is assumed that the reader possesses some basic background on this topic.
- in the material and method section, add the country of manufacture for certain reagents, especially cytokines, because the country of manufacture is not specified in that section, while the country of manufacture is specified in some other sections. Everything needs to be uniform.
Modifications suggested by the reviewer were included. Please see the countries of manufacture added in red to the materials and methods section.
- at the end of the part of the text on measuring cytokines, add a reference that shows the method of measurement and add, as previously shown for cytokine production, Multiomic analysis of cytokines in immuno-oncology. Expert Rev Proteomics. 2020 Sep;17(9):663-674.
Modifications requested (lines 244-262) were included as suggested. The reference was added as requested by the reviewer.
- the first sentence in the results and discussion section begins with the objectives of the work, which should be in the introductory part of the text, while the findings should be in the results.
Modifications suggested by the reviewer were included. Please see the revised version of the manuscript on page 2.
- so change the introduction and supplement that part with clear objectives of the work.
Modifications suggested by the reviewer were included. Please see the revised version of the manuscript on page 2.
- I think that it is much better to write the results section in the paper, as in most other papers, and especially the discussion, because that way the confusion that exists in this text is avoided.
We appreciate the reviewer suggestions. Nevertheless, after analyzing this alternative of separating results from discussion, we believe that in this paper it is more coherent to analyze results and discussion together, to help the reader comprehend our conclusions in a more straightforward way. Writing the discussion as an independent section turns out to be repetitive or hard to follow, especially considering the lectins results for a non-expert on the field. Last, but not least, this journal accepts both a separate or an integrated discussion.
- For the flow cytometry results in Figure 1b, it would have been much clearer if the cell gating strategy was attached immediately with the results and not in the supp figure.
We have included the gating strategy in Figure 1B as suggested and clarified the control condition in the legend to Figure 1. Please see modified text in red on page 3.
- In Figure 2c, 2d, it is not clearly indicated what the control is. Somewhere it says control in the text, but in the pictures, it doesn't say control in the legend, but only the name of the cells, so it's confusing.
The legends indicating correspondence to bar colors were added in Figure 2 (now Figure 3), together with a full explanation of what the figure shows in the corresponding legend (lines 153-160, page 6).
"Figure 3. Glycophenotype of Tn-bearing LL/2 and WT cells. LL/2 cells were stained with plant lectins and analyzed by flow cytometry. A) Representative gating strategy. B-H) Histogram and median fluorescence intensity (MFI) is shown for LL/2-Tn+-H12 (green) and LL/2-Tn+-F9 (pink) relative to WT LL/2 cells for VVL (B), JAC (C), HPA (D), SBA (E), DBA (F), MAL II (G) and SNA (H). Asterisks represent significant differences (*p<0.05, **p<0.01, ***p<0.001, ****p<0.00001)."
- The sentence describing the flow cytometry findings is not clearly written: Flow cytometry analysis revealed a more homogeneous recognition of LL/2-Tn+ -F9 by the anti-Tn D4 antibody as compared to LL/2-Tn+ H12 (Figure 1B), although no significant differences in the median fluorescence intensity 85 were found between the two cell variants (Figure 1C).
The paragraph was modified to present results different. Please see text in red between lines 84 and 88 in the revised version of the manuscript:
“Flow cytometry analysis revealed both Tn+ cell lines have a significant increase in Tn expression when stained with by the anti-Tn D4 antibody, compared to the parental line, being LL/2-Tn+ -F9 a more homogeneous population regarding Tn expression as compared to LL/2-Tn+ H12 (Figure 1B), although no significant differences in the median fluorescence intensity were found between the two cell variants (Figure 1C).”
- Immediately below it is written that the enzymes showed different activity without describing how and in what way a computer-drawn picture is already presented and it is not the result.
Modifications suggested by the reviewer were included. The text was modified to clarify this point. Please see text in red included between lines 84 and 91 on pages 2 and 3:
“In addition, T-synthase activity, the enzyme responsible of elongating the Tn antigen and the formation of core 1 (Figure 1D), was abrogated in both Tn+ cells (Figure 1E), thus confirming interruption of the step catalyzed by this enzyme.”
- After that, the authors write that the findings that show the difference are in the Supplementary Figure. Everything is completely confused
We acknowledge reviewer suggestion. Nevertheless, the results shown in supplementary figure 1 are regarding cellular proliferation, migration, and colony formation, not the enzyme activity. We have also included the figure in the revised manuscript to accompany the results described in the text. Please see text between lines 102 and 104 and Figure 2 on page 4.
- in the discussion regarding tumor suppression of cells in the tumor environment, cite references that show that numerous mediators and cytokines induce suppression and changes in membrane antigen expression that have been shown in various other models: a) The role of cytokines in the regulation of NK cells in the tumor environment. Cytokine. 2019 May;117:30-40. b) among many cytokines, TNF directly induces changes in the antigenic expression of individual cells, which has been demonstrated through the mechanism of tearing off tumor antigens and induction of apoptosis.
TNF-α induced apoptosis is accompanied by rapid CD30 and slower CD45 shedding from K-562 cells. J Membr Biol. 2011, 239(3):115-22.
As suggested by the reviewer, the text was modified and references added on page 10. Modified text is in red shown between lines 245 and 260. Complementary references were also added (reference nº 46 to 49).
- In the legend and results in Figure 2, the name Glycophenotype is imprecise and general, it should be changed and named professionally. Somewhere in the text, one is mentioned, in the legends another, but it is not clearly written in the essence that it is the expression of markers on the surface of the cells.
As susggested by the reviewer, the title of the legend to Figure 2 (3 in the revised version of the manusacript) was changed by “Surface glycan motifs of Tn-bearing LL/2 and WT cells determined by lectin recognition.”
- In Figure 2, the legends also do not say what the control is, but it is only assumed that the first column is the control population of cells.
The legend to Figure 2 (Figure 3 in the revised manuscript on page 6) was changed to clearly describe the control condition:
"Figure 3. Surface glycan motifs of Tn-bearing LL/2 and WT cells determined by lectin recognition. LL/2 cells were stained with plant lectins and analyzed by flow cytometry. A) Representative gating strategy. B-H) Histogram and median fluorescence intensity (MFI) is shown for LL/2-Tn+-H12 (green) and LL/2-Tn+-F9 (pink) relative to WT LL/2 cells for VVL (B), JAC (C), HPA (D), SBA (E), DBA (F), MAL II (G) and SNA (H). Dotted histograms represent the control condition consisting of cells incubated only in presence of streptavidin-FITCAsterisks represent significant differences (*p<0.05, **p<0.01, ***p<0.001, ****p<0.00001)."
- It would be good if the original flow cytometry data were first shown in Figure 2, and only then the bars with immunophenotype values
Modifications were included in the revised version of the paper as suggested by the reviewer. Please see Figure 3 in the revised manuscript.
- In the discussion and results, it is not written at all why CD86 was analyzed and for what reason and what its analysis means, but it is left to the reader to guess what the researchers wanted to say...
This was clarified in the first paragraph of section 2.3 (page 9). CD80 and CD86 were determined since they are key costimulatory molecules in the activation of T cells, while cytokines determine the differentiation of T cells.
“To analyze the capacity of Tn+ LL/2 tumor cells to modulate the maturation of DCs, we incubated BMDC (which express MGL2, Figure 5A) with soluble components derived from LL/2-Tn+-H12, LL/2-Tn+-F9 or WT cells present in the conditioning medium (CM). Then, we evaluated the expression of costimulatory molecules on BMDCs (such as MHCII, CD80 and CD86) by flow cytometry (Figure 5B) and the secretion of pro- and anti-inflammatory cytokines (IL-12 and TNFα, IL-10) by specific sandwich ELISA (Figure 5C). “
- the variation in cytokine production in Figure 4 is different and this is not explained at all, which is why the changes in the concentration of TNF, which is proinflammatory and for which a reference was previously added that should be inserted, as well as for IL10, which is an inhibitory cytokine, are only repeated findings that can be seen from the picture
As previously requested by the reviewer we have expanded/developed the discussion related to the immunomodulatory properties of Tn-bearing tumor cells. We have presented the role of the evaluated cytokines, as well as others, on pages 9 and 10, included references accordingly and explain the main obtained conclusions.
Round 2
Reviewer 2 Report
the authors wrote that they corrected the work according to the instructions but did not correct according to the instructions
Author Response
Dear Reviewer
We have recently uploaded our corrections based on your instructions. However, you returned to us telling to us that "the authors wrote that they corrected the work according to the instructions but did not correct according to the instructions" We are rather surprised since we gave a point-to-point answer to all your concerns. Could it be a problem with the online system? We paste below once again all the corrections and the revised manuscript. Please, if you are not allowed to see them, contact the editor to ask for some help. Thank you very much for your time and your helpful advice.
- the introduction is too crowded with data about glycans, and there is no significant connection between what the authors write and what is the goal of the work. Why all that was investigated is not very clearly written.???
The introduction has several paragraphs
- lung cancer and the importance of glycoantigens in its growth.
- C-type lectin receptors and MGL2
- Use of Crispr technology
- A final paragraph indicating the objective of the study and the obtained results and conclusions.
We are convinced, that the information in this section is all relevant. However, to follow the reviewer´s advice, we have included a sentence explaining the goal of the study (lines 65-66).
In this study we aimed at studying the role aberrant O-glycosylation in the interaction by MGL2 and by dendritic cells, the main orchestrator of adapting immunity. To this end,
We would also like to point out that this paper has been submitted to the special Issue “New insights on roles of glycoconjugates in Health and disease” covers topics related with glycosylation and it is assumed that the reader possesses some basic background on this topic.
- in the material and method section, add the country of manufacture for certain reagents, especially cytokines, because the country of manufacture is not specified in that section, while the country of manufacture is specified in some other sections. Everything needs to be uniform.
Modifications suggested by the reviewer were included. Please see the countries of manufacture added in red to the materials and methods section.
- at the end of the part of the text on measuring cytokines, add a reference that shows the method of measurement and add, as previously shown for cytokine production, Multiomic analysis of cytokines in immuno-oncology. Expert Rev Proteomics. 2020 Sep;17(9):663-674.
Modifications requested (lines 244-262) were included as suggested. The reference was added as requested by the reviewer.
- the first sentence in the results and discussion section begins with the objectives of the work, which should be in the introductory part of the text, while the findings should be in the results.
Modifications suggested by the reviewer were included. Please see the revised version of the manuscript on page 2.
- so change the introduction and supplement that part with clear objectives of the work.
Modifications suggested by the reviewer were included. Please see the revised version of the manuscript on page 2.
- I think that it is much better to write the results section in the paper, as in most other papers, and especially the discussion, because that way the confusion that exists in this text is avoided.
We appreciate the reviewer suggestions. Nevertheless, after analyzing this alternative of separating results from discussion, we believe that in this paper it is more coherent to analyze results and discussion together, to help the reader comprehend our conclusions in a more straightforward way. Writing the discussion as an independent section turns out to be repetitive or hard to follow, especially considering the lectins results for a non-expert on the field. Last, but not least, this journal accepts both a separate or an integrated discussion.
- For the flow cytometry results in Figure 1b, it would have been much clearer if the cell gating strategy was attached immediately with the results and not in the supp figure.
We have included the gating strategy in Figure 1B as suggested and clarified the control condition in the legend to Figure 1. Please see modified text in red on page 3.
- In Figure 2c, 2d, it is not clearly indicated what the control is. Somewhere it says control in the text, but in the pictures, it doesn't say control in the legend, but only the name of the cells, so it's confusing.
The legends indicating correspondence to bar colors were added in Figure 2 (now Figure 3), together with a full explanation of what the figure shows in the corresponding legend (lines 153-160, page 6).
Figure 3. Glycophenotype of Tn-bearing LL/2 and WT cells. LL/2 cells were stained with plant lectins and analyzed by flow cytometry. A) Representative gating strategy. B-H) Histogram and median fluorescence intensity (MFI) is shown for LL/2-Tn+-H12 (green) and LL/2-Tn+-F9 (pink) relative to WT LL/2 cells for VVL (B), JAC (C), HPA (D), SBA (E), DBA (F), MAL II (G) and SNA (H). Asterisks represent significant differences (*p<0.05, **p<0.01, ***p<0.001, ****p<0.00001).
- The sentence describing the flow cytometry findings is not clearly written: Flow cytometry analysis revealed a more homogeneous recognition of LL/2-Tn+ -F9 by the anti-Tn D4 antibody as compared to LL/2-Tn+ H12 (Figure 1B), although no significant differences in the median fluorescence intensity 85 were found between the two cell variants (Figure 1C).
The paragraph was modified to present results different. Please see text in red between lines 84 and 88 in the revised version of the manuscript:
“Flow cytometry analysis revealed both Tn+ cell lines have a significant increase in Tn expression when stained with by the anti-Tn D4 antibody, compared to the parental line, being LL/2-Tn+ -F9 a more homogeneous population regarding Tn expression as compared to LL/2-Tn+ H12 (Figure 1B), although no significant differences in the median fluorescence intensity were found between the two cell variants (Figure 1C).”
- Immediately below it is written that the enzymes showed different activity without describing how and in what way a computer-drawn picture is already presented and it is not the result.
Modifications suggested by the reviewer were included. The text was modified to clarify this point. Please see text in red included between lines 84 and 91 on pages 2 and 3:
“In addition, T-synthase activity, the enzyme responsible of elongating the Tn antigen and the formation of core 1 (Figure 1D), was abrogated in both Tn+ cells (Figure 1E), thus confirming interruption of the step catalyzed by this enzyme.”
- After that, the authors write that the findings that show the difference are in the Supplementary Figure. Everything is completely confused
We acknowledge reviewer suggestion. Nevertheless, the results shown in supplementary figure 1 are regarding cellular proliferation, migration, and colony formation, not the enzyme activity. We have also included the figure in the revised manuscript to accompany the results described in the text. Please see text between lines 102 and 104 and Figure 2 on page 4.
- in the discussion regarding tumor suppression of cells in the tumor environment, cite references that show that numerous mediators and cytokines induce suppression and changes in membrane antigen expression that have been shown in various other models:
- a) The role of cytokines in the regulation of NK cells in the tumor environment. Cytokine. 2019 May;117:30-40.
- b) among many cytokines, TNF directly induces changes in the antigenic expression of individual cells, which has been demonstrated through the mechanism of tearing off tumor antigens and induction of apoptosis.
TNF-α induced apoptosis is accompanied by rapid CD30 and slower CD45 shedding from K-562 cells. J Membr Biol. 2011, 239(3):115-22.
As suggested by the reviewer, the text was modified and references added on page 10. Modified text is in red shown between lines 245 and 260. Complementary references were also added (reference nº 46 to 49).
- In the legend and results in Figure 2, the name Glycophenotype is imprecise and general, it should be changed and named professionally. Somewhere in the text, one is mentioned, in the legends another, but it is not clearly written in the essence that it is the expression of markers on the surface of the cells.
As susggested by the reviewer, the title of the legend to Figure 2 (3 in the revised version of the manusacript) was changed by “Surface glycan motifs of Tn-bearing LL/2 and WT cells determined by lectin recognition.”
- In Figure 2, the legends also do not say what the control is, but it is only assumed that the first column is the control population of cells.
The legend to Figure 2 (Figure 3 in the revised manuscript on page 6) was changed to clearly describe the control condition:
Figure 3. Surface glycan motifs of Tn-bearing LL/2 and WT cells determined by lectin recognition. LL/2 cells were stained with plant lectins and analyzed by flow cytometry. A) Representative gating strategy. B-H) Histogram and median fluorescence intensity (MFI) is shown for LL/2-Tn+-H12 (green) and LL/2-Tn+-F9 (pink) relative to WT LL/2 cells for VVL (B), JAC (C), HPA (D), SBA (E), DBA (F), MAL II (G) and SNA (H). Dotted histograms represent the control condition consisting of cells incubated only in presence of streptavidin-FITCAsterisks represent significant differences (*p<0.05, **p<0.01, ***p<0.001, ****p<0.00001).
- It would be good if the original flow cytometry data were first shown in Figure 2, and only then the bars with immunophenotype values
Modifications were included in the revised version of the paper as suggested by the reviewer. Please see Figure 3 in the revised manuscript.
- In the discussion and results, it is not written at all why CD86 was analyzed and for what reason and what its analysis means, but it is left to the reader to guess what the researchers wanted to say...
This was clarified in the first paragraph of section 2.3 (page 9). CD80 and CD86 were determined since they are key costimulatory molecules in the activation of T cells, while cytokines determine the differentiation of T cells.
“To analyze the capacity of Tn+ LL/2 tumor cells to modulate the maturation of DCs, we incubated BMDC (which express MGL2, Figure 5A) with soluble components derived from LL/2-Tn+-H12, LL/2-Tn+-F9 or WT cells present in the conditioning medium (CM). Then, we evaluated the expression of costimulatory molecules on BMDCs (such as MHCII, CD80 and CD86) by flow cytometry (Figure 5B) and the secretion of pro- and anti-inflammatory cytokines (IL-12 and TNFα, IL-10) by specific sandwich ELISA (Figure 5C). “
- the variation in cytokine production in Figure 4 is different and this is not explained at all, which is why the changes in the concentration of TNF, which is proinflammatory and for which a reference was previously added that should be inserted, as well as for IL10, which is an inhibitory cytokine, are only repeated findings that can be seen from the picture
As previously requested by the reviewer we have expanded/developed the discussion related to the immunomodulatory properties of Tn-bearing tumor cells. We have presented the role of the evaluated cytokines, as well as others, on pages 9 and 10, included references accordingly and explain the main obtained conclusions.
